# Bicontinuous Interfacially Jammed Emulsion Gels (Bijels): Preparation, Control Strategies, and Derived Porous Materials

**DOI:** 10.3390/nano14070574

**Published:** 2024-03-26

**Authors:** Xingliang Shen, Meiwen Cao

**Affiliations:** State Key Laboratory of Heavy Oil Processing, Department of Biological and Energy Chemical Engineering, College of Chemical Engineering, China University of Petroleum (East China), 66 Changjiang West Road, Qingdao 266580, China; shenxingliang123@163.com

**Keywords:** Bijels, preparation methods, control strategies, porous materials

## Abstract

Bicontinuous interfacially jammed emulsion gels, also known as Bijels, are a new type of soft condensed matter. Over the last decade, Bijels have attracted considerable attention because of their unique morphology, property, and broad application prospects. In the present review, we summarize the preparation methods and main control strategies of Bijels, focusing on the research progress and application of Bijels as templates for porous materials preparation in recent years. The potential future directions and applications of Bijels are also envisaged.

## 1. Introduction

The liquid–liquid interface stabilizer of colloid particles originates from the Pickering emulsion, which was discovered in the early 20th century by Ramsden [1] and Pickering [2]. Compared with traditional emulsions stabilized by surfactants, the solid film formed by the self-assembly of colloidal particles at the oil–water interface can protect the microdroplets against coalescence and Ostwald ripening [3,4]. According to the thermodynamic model established by Pieranski [5], for a single spherical solid particle with an effective radius of r, the reduction in the system free energy is the main driving force for particle assembly at the liquid–liquid interface. The interface free energy Δ*E* can be expressed by the following equation [6]:(1)$$∆E=\pi r^{2}\gamma_{o/w}{\left(1–\left|cos\theta \right|\right)}^{2}$$
where *r* is the particle radius, *γ*_o/w_ is the liquid–liquid interfacial tension, and *θ* is the three-phase contact angle of the particle. For micron-scale colloidal particles, the free energy of the system is reduced much more than the heat energy *k*_B_*T* (*k*_B_*T* is the heat energy of particles leaving the interface; *k*_B_ is the Boltzmann constant; and *T* is the absolute temperature), so the particles can be stably adsorbed at the interface [6]. For a 250 nm radius particle at a water–lutidine interface, the interface free energy at 40 °C is 10^4^ *k*_B_*T* [7]. However, because the dispersed phase in Pickering emulsions is spherical in structure, with the lowest surface-to-volume ratio and isolated, it is limited in terms of fluid–fluid interface transport and catalytic reaction. Maximizing the interfacial area of Pickering emulsions is advantageous for applications, such as energy storage and conversion [8,9,10], gas adsorption and separation [11], catalyst carriers [12], and air purification [13,14,15].

The idea of stabilizing the bicontinuous liquid–liquid interface with interfacially jammed nanoparticles dates back to the work of Cates et al. [16] in 2005. They used the lattice Boltzmann method to explore a kinetic pathway that leads to the creation of a bicontinuous interfacially jammed emulsion gel (Bijel), which was experimentally confirmed by Clegg et al. [17] two years later. For the formation of the Bijel structure, colloidal particles with moderate wettability were introduced into two incompatible liquids, typically oil and aqueous phases. In the process of phase separation undergoing spinodal decomposition, the newly generated liquid–liquid interface binds and adsorbs colloidal particles. Finally, a pair of interpenetrating, bicontinuous fluid domains are solidified into a stable arrangement by densely jammed colloidal particles at the fluid–fluid interface [18].

Bijels were first experimentally prepared via thermally induced phase separation (TIPS) in the presence of nanoparticles [17], which subsequently inspired researchers to develop fabrication approaches including solvent transfer-induced phase separation (STRIPS) [19], vaporization-induced phase separation (VIPS) [20], and direct mixing [21]. In the initial Bijel studies, thermally induced phase separation was widely used. Thermally induced demixing of initially miscible fluids provided a path for the formation of different fluid–fluid interfacial arrangements. The low molecular weight liquid system of water and 2,6-lutidine with the lowest critical solution temperature (LCST) was a typical system used for the construction of Bijels [22]. As an alternative to VIPS, STRIPS is a continuous process used to produce Bijels without using experimentally delicate parameters, such as a specific temperature and solvent [23]. Two immiscible liquids are dissolved in the third phase cosolvent to form a ternary mixture. This ternary mixture allows for the dispersion of colloidal particles via fine tuning of the particle surface chemistry. Then, the mixture is injected into the water phase flowing in the same direction through a specific channel. With diffusion of the cosolvent to the water phase, the spinodal decomposition is triggered, providing a powerful platform to produce Bijels.

However, the volume of materials that can be produced by STRIPS is limited, with production time per unit volume of materials increasing rapidly as channel width decreases, which directly restricts its industrial application. Remarkably, Bijel-like morphologies can be generated simply by shaking or mixing immiscible liquids in the presence of nanoparticles [24]. In short, each method of Bijel fabrication has its own advantages and disadvantages, while the prepared structures have different characteristics (e.g., final morphology and domain size). It is necessary to choose the appropriate method to prepare suitable Bijels for specific applications.

It is known that the properties of Bijels are crucially dependent on the colloidal particles, including their surface wettability, size, concentration, and charge properties [25,26,27]. The interfacial self-assembly of colloidal particles at a liquid–liquid interface due to liquid favorable interactions minimizes the free energy of the system and leads to the formation of jammed structures [5,28]. It is worth emphasizing that equal wettability of the particle surfaces by the two liquid phases (neutral wetting) is the key to the formation of Bijels [29]. When the particle wettability deviates from neutral, droplets are more likely to be formed. In particular, the fluid bicontinuous structure can be systematically varied by modifying the property of the colloidal particles [30,31]. Thus, when Bijels are used as templates, it is possible to prepare structures with specific morphologies by adjusting the surface properties of the particles to fulfill the material requirements for various applications [32].

Determining how to design and prepare multistage porous materials with a rich structure, excellent performance, and wide applications is still a key point in this field [9,10,33]. As a new method for preparing porous materials, the Bijel templates show obvious advantages [34,35,36]: (1) the preparation process does not need to introduce surfactants; (2) the interfacial layer composed of colloidal particles is more stable and rigid; (3) the microstructure of porous materials can be regulated by changing the wettability, concentration, and charge property of the colloidal particles; and (4) the porous materials prepared using the method have uniform pore channels, narrow pore size distribution, and large pore size (nanometer to micrometer). The simultaneous presence of interconnected polar and nonpolar domains makes these porous materials suitable for various applications in the fields of molecular encapsulation [25,37], tissue engineering [38], drug delivery [39], and electrochemistry [40].

Herein, the main methods of Bijel preparation in the laboratory are shown in the second section. The third section focuses on strategies to control the Bijel structure. Finally, the research achievements and breakthrough progress of Bijels in recent years are discussed and the future research directions of Bijels are proposed.

## 2. Preparation Methods of Bijels

Bijels were originally produced by TIPS of the water and 2,6-lutidine mixture [17]. STRIPS, as a complementary method, makes use of a cosolvent to induce mixing/demixing at room temperature and allows for a broad selection of oils to produce Bijels in various forms. Recently, Bijels have been prepared by simple mixing of a surfactant-doped mixture of two immiscible fluids loaded with colloidal particles.

### 2.1. TIPS

The Bijel structures were first generated from two low molecular weight liquids using the spinodal decomposition technique [16]. During Bijel formation, a population of particles is dispersed at the critical liquid composition in the mixed phase. TIPS via spinodal decomposition is induced by a change in temperature, leading to the particles to be swept up and irreversibly trapped in the newly created liquid–liquid interface [18].

It should be noted that the symmetry of the phase diagram is also an important consideration in Bijel fabrication [41]. Bijels are quenched into the spinodal region of the liquid–liquid phase diagram using a rapid change in temperature. A binodal line (solid line) separates a mixed phase from a region where the liquids are phase separated in the phase diagram (Figure 1a) [42]. The kinetic pathway associated with phase separation depends on the proportion of the two liquids and the degree of temperature change [41]. There is a spinodal line (dotted line) inside the demixed region, which is the boundary between the totally unstable mixed phase and the relatively stable mixed phase. The spinodal line meets the binodal line at the critical point. Spinodal decomposition can be achieved either by passing through the critical point or via a deep and fast quench at a different composition (Figure 1a). A small change in temperature, which only takes the system across the binodal line, results in phase separation via nucleation [42] and growth [23,43]. After nucleation separation, a liquid (usually with a small volume fraction) often exists in the form of spherical droplets to reduce the interface volume between the two liquids, so nucleation separation usually produces only independent spherical droplets.

For simple binary systems, quenching (referring to rapid heating or cooling) of the two-phase region triggers the spinodal decomposition [45]. A temperature change, which takes partially miscible liquids from the mixed phase across both the binodal and the spinodal lines, results in phase separation via spinodal decomposition (Figure 1a). In the process of phase separation, colloidal particles are enriched at the phase interface as a stabilizer, which makes the phase structure stable in the bicontinuous state. Typically, Bijel structures produced by TIPS are constructed using the solvent system of water and 2,6-lutidine [34]. When the volume fraction of 2,6-lutidine was 6.4%, the LCST of the system was 34.1 °C. It was found that the spinodal decomposition occurred on the whole depth of the sample, and the resulting structure was fully three-dimensional bicontinuous. Moreover, the Bijel structure can be formed in an ethylene glycol and nitromethane system via TIPS [41]. Compared with the water and 2,6-lutidine system, the Bijel structure prepared by ethylene glycol and nitromethane system had an upper critical solution temperature (≈40 °C).

The three-dimensional bicontinuous structure generated by spinodal decomposition is useful in many applications [18], while the zero-mean curvature of the Bijels makes them resistant to Ostwald ripening [46]. If well controlled, it can produce liquid bicontinuous systems with macroscopically length-scale permeable domains with single, well-defined, controllable channel widths that may be down to the microscale [22]. If a specific uniform quenching can be achieved, large quantities (on the order of milliliters) of Bijel samples can be produced using the TIPS method. Although scaling is difficult using this approach, it may enable the use of Bijels in high value-added applications. However, the number of liquid pairs that can show this phase behavior for TIPS is limited in the attainable temperature range (e.g., 25 °C to 90 °C), restricting the palette of fluids from which Bijels can be made. Meanwhile, the solvent used in this method (such as 2,6-lutidine or nitromethane) is a low-toxicity but harmful substance. There is a risk of explosion in the event of high temperature, so this system has greater limitations in biological applications. Furthermore, the difficulty of TIPS lies in even quenching of the system to achieve demixing of the two liquids.

### 2.2. STRIPS

STRIPS has also been proven to be an effective way to prepare Bijel structures. The two phases of oil and water are dissolved in the third phase cosolvent (such as ethanol) to form a ternary homogeneous mixture. The rapid loss or diffusion of the cosolvent causes the system to shift below the spinodal line, producing a water-rich and oil-rich phase. The removal or transfer of the solvent from the ternary mixture is the main factor triggering phase separation [32]. Generally, a coaxial extrusion device composed of two centered glass capillaries is connected via tubing to syringe pumps that enable the continuous flow of both the ternary mixture and water. The ternary homogeneous mixture of oil–water cosolvent is injected into the water phase flowing in the same direction [44]. During this process, solvent diffused to the water stream and demixing take place within the initial homogenous mixture. The spinodal decomposition of the two-phase fluid is triggered, resulting in the Bijel structure (Figure 1b).

For TIPS, regardless of how the temperature quenching is performed or how the particle wetting is manipulated, it is extremely challenging to adjust the resulting geometry of the Bijels produced by spinodal decomposition. In contrast to TIPS, the continuous production route of Bijels in the form of fibers, planar films, and microparticles is unique to STRIPS. STRIPS, as a temperature-free technique, has developed rapidly in the past decade for the preparation of Bijels with different morphological structures at microscale [33]. This method also had a wide range of oil phase options, which greatly optimizes the preparation process of Bijels. For example, a hollow fiber nanocomposite membrane with bicontinuous morphology can be prepared by STRIPS using a photopolymerization monomer (1-6 hexadiol diacrylate) as the oil phase [47]. Furthermore, STRIPS Bijels have asymmetric architectures, with small bicontinuous surface domains and larger internal domains. The unique structures of STRIPS Bijels show great opportunities as catalytic microreactors for the continuous production of specialty chemicals, pharmaceuticals, and biofuels [48].

However, using STRIPS to form Bijels has intrinsic disadvantages. The solvent environment in the ternary liquid mixture is not conducive to the dispersion of silica particles [49]. This can be related to incompatibility between the hydrophilic silica surface and the hydrophobic constituents of the ternary mixture. In order to achieve dispersion, silica particles must be modified in situ. While there have been some examples of bypassing this challenge by in situ cationic surfactant activation [50] or siloxane modification [51], achieving finely tuned particle wetting remains a significant challenge. Because the process of the STRIPS method is complicated and contains many flammable components, the wide application of this method is limited [52]. Furthermore, the role of the ternary liquid phase equilibrium and the partitioning of the solvent between the phases requires further investigation. More research is also needed to connect the STRIPS Bijel structure with the nanoparticle surface chemistry.

### 2.3. Direct Mixing

In the last decade, researchers have worked to develop methods to produce liquid bicontinuous structures using interfacial jamming of particles without using spinodal decomposition. A highly desirable goal is to produce a material with a similar morphology to that of the tortuous Bijels produced via spinodal decomposition using a method that is less challenging to work with. Direct mixing is a simple and efficient method for preparing Bijels. The combination of nanoparticles’ and polymers’ complementary functionality can be used to easily prepare Bijels [24]. The carboxylic acid-functionalized polystyrene (PS-COOH) nanoparticles dispersed in water and the amine-functionalized polydimethylsiloxane (PDMS-NH_2_) in oil stabilizes the liquid–liquid fluid interface to form an elastic nanoparticle film. Since the liquid bicontinuous system is produced under a high shear rate, it consists of droplets during the initial formation stage. After homogenization by a vortex oscillator, a jamming interface film composed of nanoparticles and surfactants is formed at the oil–water interface (Figure 1c). The Bijels prepared by this method have a submicron internal size (up to 300 nm), which is an order of magnitude lower than the minimum results reported by traditional methods.

An alternative strategy for generating Bijel structures was developed by combining interfacial nanoparticles and molecular surfactants with highly viscous immiscible solutions [21]. The Bijel structure is obtained by direct mixing at room temperature in a high viscosity glycerine and silicone oil system. The high viscosity system produces a brief bicontinuous phase state during mixing. It was also found that the smaller the density difference between the two phases, the more stable the Bijel structure. The two liquids are connected in three dimensions, indicating that the direct agitation method can form an ideal Bijels structure. The stirring protocol was found to be critical to the generation of bicontinuous structures.

Obviously, direct mixing avoids the defects of traditional preparation methods and greatly simplifies the preparation steps of Bijels. Production via spinodal decomposition has proven difficult to scale beyond milliliter-level production [23]. Batch shaking and stirring may provide a pathway to increasing scale for greater industrial applications. Bijels prepared by direct mixing have potential applications in multiphase microreactors, microfluidic devices, membrane contactors, and multiscale porous materials [24]. Unsurprisingly, however, there are many structural differences between systems produced by direct mixing and those produced by spinor decomposition. Bijels produced by direct mixing have been reported to lack the presence of major length scales that can be characterized by Fourier transform of confocal micrographs. Meanwhile, the extent of the three-dimensional continuous structure in these systems is unclear. The actual mechanism by which these structures are formed, and how they might be controlled, are still not clearly known.

Additionally, the vaporization-induced phase separation (VIPS) [10,53], the centrifugal method [54], and the phase inversion method [55] have also been proven to be effective to form Bijel structures. For VIPS, phase separation is triggered by vaporization of a volatile cosolvent from a homogeneous mixture into air. The ternary mixture formulation follows the similar principles as that of STRIPS, so the VIPS method is not further outlined here. The other two methods are in the very early stage of research, so are also not discussed extensively here.

## 3. Control Strategies of Bijels

Each preparation method has its own factors that affect the formation of the Bijels structure [56,57]. For example, in the solvent transfer method, it is possible to tune Bijels formation via changes in parameters such as composition, quench depth, and quench rate [21]. As the stabilizer of the two-phase interface, colloidal particles are the general control factor of different methods. Hence, understanding and controlling the properties of colloidal particles is central to the creation of Bijels. The morphology and domain size of the Bijels are closely related to the surface wettability, concentration, size, and charge properties of particles [53].

### 3.1. Surface Wettability of Particles

As particles are pushed close to one another at a liquid–liquid interface, the particle wettability tends to bias the liquid interface towards adopting a particular curvature [29]. To yield Bijels, the nanoparticles need to have the same wettability in both liquid phases. That is, the three-phase contact angle (*θ*) of the particles should be close to 90° [43]. The actual *θ* is one of the most difficult factors to predict and control (or even to measure), even for perfectly spherical particles [58]. Fortunately, direct or indirect methods of accurately measuring the wettability have been explored in several works [59,60,61].

In the initial studies, a systematic drying procedure was used to temporarily tune the wettability of the particles [62]. The original particles of silica used in the Bijel templating usually have a layer of physically adsorbed water on the surface. The water layer has a profound effect on the wettability of the particles. At about 25 °C, the silica surface is covered with a complete layer of physically adsorbed water and there is no modification to the chemistry of the silanol groups. The silanol groups remain unmodified up to about 190 °C; however, the adsorbed layer of water is driven off by the time this temperature is reached [63]. A standard and effective way to achieve neutral wetting is chemical surface modification. For silica particles used in many types of Bijel templating, the silanization treatment replaces relatively hydrophilic Si-OH (silanol) groups on their surface. With the use of hexalmethyldisilazane (HMDS), the wetting property of silica colloids can be modified to fabricate a novel Bijel system comprising the ethanediol–nitromethane binary liquid [25].

Alternatively, due to the complicated steps and high cost of surface treatment of the particles [64], most experiments are prepared using cetyltrimethyl ammonium bromide (CTAB) to adjust the wetting properties of silica nanoparticles in situ (especially for the STRIPS method). Upon increasing the CTAB concentration, the morphologies of Bijel fibers changed from separated voids to cavernous interconnected voids with domain sizes of several hundred nanometers (Figure 2a) [32]. The increase in CTAB concentration may render the nanoparticles more hydrophobic, resulting in smaller domain sizes. Surfactants are used to modify nanoparticle wetting in situ to facilitate the attachment and jamming of nanoparticles at the oil–water interface, which is an essential step in the formation of Bijels. However, the introduction of surfactants has restricted the biological application of Bijels [65]. Similar effects of an identical origin were observed in a combination of nanoparticles of different wetting without adding any surfactant (Figure 2b) [66]. By adjusting the ratio of the two nanoparticles, neutral wetting of heterogeneous clusters can be achieved, thus forming a bicontinuous morphology. This method simplifies production, enhances reproducibility, and is an important step for scaling up Bijels for industrial applications.

### 3.2. Particle Concentration

The size of the fluid domain can be controlled by changing the concentration of the colloidal particles. In general, the larger the volume fraction of colloidal particles, the smaller the obtained Bijel domain size. For example, when Bijels were prepared using the TIPS method in a binary mixture of styrene trimer/polybutene (PS/PB), the size of the domain could be reduced by increasing the particle concentration [26]. At the low concentration of fluorescence hydrophobic silica nanoparticles (B-SNPs) (0.5 wt%), the bicontinuous structure was stabilized at a large domain size. An obvious domain size reduction was observed when the weight fraction of particles was increased to 2.0 wt% and 4.5 wt% (Figure 2c). Similarly, the domain size of the composite electrode was adjusted by changing the silica particle volume fraction in the range of 0.014~0.040 in a binary mixture of 2,6-lutidine and water [40]. The resulting morphology was a highly porous nickel shell with uniform, continuous pores in the range of 8~22 μm (Figure 2d).

Furthermore, the morphology and domain size of Bijels prepared by the STRIPS method can also be tailored by selection of the particle concentration. With the increase in the silica fraction from 0.9 to 8.7 vol%, the morphology of Bijels changed significantly. The morphology features changed from initial separated voids and shapes, to a continuous channel extending throughout the fiber (Figure 2e). Domain sizes of Bijel fibers were less than 2 microns when the particle concentration was increased (Figure 2a). Furthermore, the particle concentration is crucial for the formation of the Bijel structure using the direct mixing method [24]. A particle concentration that is too low is not conducive to the formation of the solidification interface during the spinodal decomposition process to prevent further phase separation, resulting in instability of the system. Only when the particle concentration is increased to a specific concentration (0.5 mg ml^–1^) can a stable Bijel structure be formed.

### 3.3. Particles Size

It can be concluded from Equation (1) that, besides the wettability, the size of particles also has a profound effect on their desorption energy. It seems that the desorption energy of colloidal particles at the oil–water interface increases with the increase in the particle size; however, a particle size that is too large is unfavorable to the initial adsorption of particles [67]. The adsorption kinetics of the larger particles are slow and result in high adsorption barriers and less efficient packing at the interface [68]. Furthermore, the smaller particle size more easily forms a dense solid film at the fluid–fluid interface to improve the stability of the system [69]. Cate et al. [22] experimentally explored the effect of particle size on Bijel formation. It was found that Bijels prepared by the TIPS method were more robust when nanoparticles were used instead of microparticles. The minimum heating rate allowed by the nanoparticle system was two orders of magnitude slower than that of the microparticle system. The microparticle system was more likely to stabilize the emulsions at a slow rate rather than Bijels. This phenomenon was explained as follows: the interfacial microparticles with non-neutral wetting are not conducive to the bicontinuous interfacial curvature, while nanoparticles with similar wetting benefit from the mechanical-leeway mechanism [22].

It is worth emphasizing that the domain size for the final Bijels can also be controlled via the size of the colloid particles [34]. This is because the interface is trapped in a configuration that has just enough area to hold all the trapped particles, that is, fixing a specific quantity of small-sized particles will require a larger interface area [21]. As the interfacial area increases, the liquid domain size becomes correspondingly diminished. The scaling relationship between these quantities has been demonstrated from a few microns to hundreds of microns [25,69].

### 3.4. Charge Property of Particles

By changing the pH of the system, adding ionic surfactant, or applying an electric field, the charge properties of the surface of the colloidal particles may be changed, thus affecting the Bijel structure [8,64,70]. In situ modification of silica nanoparticles by electrostatic interaction with cationic surfactants has been proven to be an effective method to modify the silica surface. This method can not only be used to adjust the wettability of particles, but also to change the surface charge of particles [56]. The amount of adsorbed surfactant needs to be controlled to promote compatibility between the silica particles and the ternary mixture. In addition, a sufficiently high cation concentration causes a surface charge reversal [8].

The in situ modified silica nanoparticles can be dispersed in a Bijel mixture composed of isopropyl alcohol (IPA), diethyl phthalate (DEP), and water to form the STRIPS Bijels [71]. The adsorption of di-decyl dimethyl ammonium bromide (di-C_10_TAB) on the surface of negatively charged silica increases their surface charge density. High concentrations of di-C_10_TAB can lead to agglomeration of nanoparticles. Possibly due to the strong agglomeration of silica particles in the ternary mixture, the Bijel structure cannot be formed. Therefore, it is crucial to slow down the adsorption of di-C_10_TAB by the particles’ surface for particle dispersibility and Bijel formation.

If the surface of a particle contains functional groups that are responsive to pH, pH can regulate its surface charge properties. The protonation/deprotonation degree of the carboxyl group and amino group can be effectively adjusted by changing the pH value of solution, and then the interfacial activity of nanoparticles can be regulated [72,73]. Interestingly, in the process of Bijel preparation using thermally induced phase separation, the zeta potential of the particles is affected by different concentrations of dye labels [19]. With an increase in the fluorescein isothiocyanate (FITC) concentration, the surface potential of particles tends to become less negative in water, thus showing a preference for water-rich phase. This may be related to the increase in the particles’ preference for low dielectric constant solvents.

Notably, using nanoparticles dispersed in water and amine end-capped polymers in oil, surfactant-like nanoparticles were generated in situ at the interface, overcoming the inherent weak forces that control adsorption at the nanoparticle interface. The droplet morphology was controlled by electric field force [64].

In short, the adjustability of the Bijel structure undoubtedly increases its application potential. The morphology and domain size of Bijels is tunable in a range relevant to a number of important applications simply through altering the property of particles. For instance, in tissue-engineering scaffolds, accessible through Bijel processing, the pore size can be large enough to accommodate cell migration, proliferation, and organization into functional tissues [74]. Similarly, converting Bijels into a co-continuous functional composite for energy applications requires the continuous deposition of several layers of materials on the inner surface of the porous scaffold and specific pore sizes [40].

## 4. Bijel-Derived Porous Materials

Porous materials are currently of great scientific and technological interest. Bijels can be used as a unique class of soft templates to synthesize functional porous materials. Their two-phase composition is independent and continuous in the materials. By solidifying a single fluid phase, Bijels have been proposed as an ideal platform for various applications including the bicontinuous porous materials, electrode porous materials, and bio-based porous materials.

### 4.1. Bicontinuous Porous Material

Microstructure analysis of the Bijel-derived materials confirmed that the key features of the parent Bijels, in particular the tight domain size distribution and the spinal-like inner surface, is preserved in different processing modes [34]. Therefore, advantages inherent to these features translate to the processed porous material. The porous materials synthesized by the Bijel templating has a uniform and inter-connective pore structure, narrow pore size distribution, high porosity and specific surface area, and large pore size range (nanometer to micrometer sized pores). By selectively introducing monomers into one of the phases of Bijels, polymer materials with a double-continuous porous structure are prepared. A new double-continuous multiporous materials group can be constructed using this material as a template.

A general platform can be developed for synthesizing porous materials with different chemical properties and tunable bicontinuous morphology using Bijel templates [37]. Firstly, Bijel templating was prepared by the TIPS method using the modified silica particles. A small amount of hydrophobic monomer and photo-initiator were added into the system, and the acrylate monomer entered the dimethylpyridine phase. Then, the polymer Bijel template was obtained by selective polymerization under ultraviolet irradiation. Using the chemical transformations of this polymer Bijel templating, a macroporous ceramic, a copper-coated macroporous polymer, a nickel network, and a spinodal nickel shell have been further prepared.

The shape of continuous porous materials is variable and adjustable [32]. It was demonstrated that the STRIPS method can produce three types of Bijel structures, namely microparticles, fibers, and membranes, through different molding processes (Figure 3a). The continuous porous materials with different shapes of Bijel structure were obtained after the oil phase was cured by photo-initiated radical polymerization. The confocal Z-stack of the fibers revealed that the STRIPS Bijel fibers had an asymmetric morphology. A gradient in the radial pore size was the unique feature of STRIPS Bijels, where the surface domain size was significantly smaller than the interior domain size.

Furthermore, the decoration of porous membranes with a dense layer of nanoparticles imparts useful functionality and can enhance membrane separation and anti-fouling properties [47]. Choosing different monomers can introduce more characteristics to the STRIPS membranes; for example, hexadiol diacrylate or butanediol diacrylate produced a highly cross-linked polymer skeleton that allows the membrane to remain stable in organic solvents. The structural clarity and asymmetry of the STRIPS membranes facilitated the control of membrane flux and selectivity, which opened potential applications for organic solvent filtration. Furthermore, Roll-to-Roll processing (R2R) enabled the fabrication of size-controlled Bijel membranes at a rate of several cubic centimeters per minute, facilitating the exciting application potential of Bijels [75].

### 4.2. Electrode Porous Material

By nature, a myriad of applications involving electrochemistry rely on the transport of different chemical species, such as analytes, ions, and electrons, to and from reactive surfaces that perform the necessary redox reactions [76,77]. Typically, these chemical species must be supplied or harvested through different phases, such as an ion- or electron-conducting materials or analyte solution. Additionally, the area of redox-reactive surfaces must typically be maximized to enhance performance. The regular arrangement of different phases, co-continuous arrangement, and controllable interface area can greatly improve the efficiency of electrochemical applications. These properties are uniquely accommodated in Bijel-derived materials because of the inherent uniformity, tunability, and bicontinuity of their template [16,40].

Inherent to the spinodal decomposition mechanism by which Bijels is formed, the fluid phases enjoy approximately uniform characteristic domain sizes. Witt et al. [40] explored the application of Bijels as a soft matter template in the synthesis of three-dimensional bicontinual composite electrodes for electrochemical energy storage and conversion. Due to the non-sequestered nature of the material phases, the bicontinuous framework can offer advanced electrochemical properties.

As an example, the Ni/Ni(OH)_2_ composite electrodes were synthesized by simple chemical processing of Bijel templating (Figure 3b) [40]. Firstly, Bijels were prepared by spinodal decomposition in a binary mixture of 2,6-lutidine and water stabilized by nanoparticle colloidal silica. The Bijel templating was then converted into a bicontinuous polymer scaffold, and nickel was electrostatically deposited on the polymer scaffold to form a 1 μm thick coating on the surface of the polymer. The coated sample was sintered in air at 500 °C to completely remove the polymer and subsequently sintered at 450 °C H_2_ (4% Ar content) to ensure the formation of pure nickel. Surprisingly, the energy and power densities can be regulated normally over a wide range, resulting in at least 1.5 times higher energy densities than those previously reported. Additionally, because of the unique topological characteristics of Bijel structures, microstructures derived from Bijel templating can enable a more uniform current density within the electrode and mitigate the loss mechanisms associated with material redistribution [77].

### 4.3. Bio-Based Porous Materials

Bio-based porous materials are commonly used in the medical field, such as in tissue-engineering scaffolds for regenerating damaged tissues and delivery systems for the controlled release of drugs [25,78]. In scaffold-based tissue engineering, biodegradable polymeric scaffolds with highly interconnected pores can facilitate cell migration and proliferation, and also nutrient transport [79]. Considering that porous materials have a stable and uniform porous structure, adjustable pore size, increased specific surface area, and surface properties, many studies have endeavored to develop porous biopolymers as controlled drug delivery matrices [80].

Bijels offer a robust, self-assembly-based platform for synthesizing a new class of morphologically unique biomaterials. Bio-based porous materials prepared by Bijel-derived templates have the advantages of unmodified, naturally encapsulated hydrogels, which can aid in overcoming drawbacks experienced in current cell delivery strategies. Typically, Bijel templating was used to generate polyethylene glycol hydrogels with a unique structure that can support fibrin, called composite Bijel-templated hydrogels (CBiTHs) [74]. Fibrin was uniformly loaded in the Bijel template’s PEG scaffold to form CBiTHs composed of PEG and fibrin phases. Then, normal human dermal fibroblasts (NHDFs) were added to fibrin/thrombin and cultured on the Bijel PEG scaffold. After 8 days, NHDFs loaded within hydrogel composite materials were able to migrate to the surrounding fibrin gels, demonstrating the viability of CBiTHs as a cell delivery system (Figure 3c).

Moreover, Bijels have attracted increasing interest as biomaterials for controlled drug delivery due to their biphasic nature in mesoscopic tortuous domains. The microstructure of this bicontinuous interface and the surface properties of nanoparticles play a key role in the mass transfer of small molecules such as active ingredients [25]. Bijel structures can release their contained matter in response to changes in temperature and solvency, and hence they show potential for controlled release applications. In vitro release studies have shown that the drugs (ethosuximide and dimethyl fumarate) can be sustainably released from the Bijel structure, following the Fick diffusion mechanism. The non-covalent drug–nanoparticle interaction changes the amount of drug released over time [39].

It is worth emphasizing that Bijels formed by phase separation of biomacromolecules (biopolymers) in water–water (W/W) systems is desirable for potential applications requiring biocompatibility [81]. In a gelatin-starch W/W system, it was demonstrated that amphoteric polystyrene latex particles were trapped at the interface and inhibited a coarser structure above the temperature of the gelatin gel, thus forming a Bijel-type morphology [82]. Phase separation in polysaccharide–polysaccharide W/W systems can be strongly influenced by the addition of a low volume fraction of silica nanoparticles [83]. The Bijel-like structures produced by this system were stable for many months. However, the design of biopolymer-based Bijels is fraught with difficulties (e.g., the extremely low interfacial tensions and maintaining neutral wetting with biopolymers). Further systematic studies of the formation of Bijel systems based on the phase separation of biological macromolecules may allow greater exploitation of such systems in novel and useful bio-based materials in the future (Table 1).

## 5. Conclusions and Outlook

Bijels are formed by networks of interpenetrating domains of two immiscible liquids stabilized by nanoparticles jammed at the fluid–fluid interface. Their tunable morphology and chemistry may find important applications in nanocasting composites, tissue engineering, electrochemistry, and fundamental studies of transport through porous media. Bijels were first reported in simulation, which subsequently spurred researchers to develop fabrication approaches including thermal quenching, cosolvent removal, and direct mixing. An inexpensive, continuous, and scalable Bijel manufacturing process has to be developed in order to promote large-scale industrial applications. At present, the process of preparing Bijels by direct mixing has been greatly simplified, but the three-dimensional property of the bicontinuous structure needs further experimental verification.

It is essential to further develop methods for preparing Bijels from nanoparticles with unique catalytic, magnetic, electronic, and photonic functions. Most Bijels, regardless of the method of fabrication, have been stabilized by silica nanoparticles. However, the interface activity of Bijel templating produced by silica is relatively limited due to its chemical inertness. Thus, the development of Bijel templating formed by metal oxides such as titanium dioxide and manganese dioxide has great potential in interfacial catalysis and double mass transfer. Furthermore, the use of naturalistic materials to manufacture Bijels will accelerate their application in the biological and medical fields, while minimizing the adverse impact of Bijels on the environment. By stabilizing the interface with biocompatible particles, such as cellulose nanocrystals, polysaccharide-based particles, and protein particles, the potential for biological applications of Bijels can be further stimulated.

## Figures and Tables

**Figure 1 nanomaterials-14-00574-f001:**
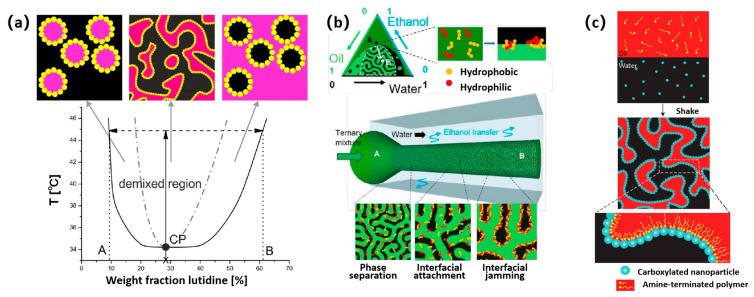
Schematic illustrations of Bijel structures prepared by different methods. (**a**) Schematic of the water-lutidine phase diagram. The solid line is the binodal, the dot–dash line is the spinodal and CP indicates the critical point. Above the binodal line, the liquids phase separate. Reprinted with permission from Ref. [42]. Copyright 2020, Royal Society of Chemistry. (**b**) Schematic diagram of the process of preparing Bijels by solvent transfer-induced phase separation. Reprinted with permission from Ref. [44]. Copyright 2020, Royal Society of Chemistry. (**c**) Schematic showing the formation of Bijels formed by the jamming of nanoparticle surfactants at the oil–water interface. Reprinted with permission from Ref. [24]. Copyright 2017, Nature Research.

**Figure 2 nanomaterials-14-00574-f002:**
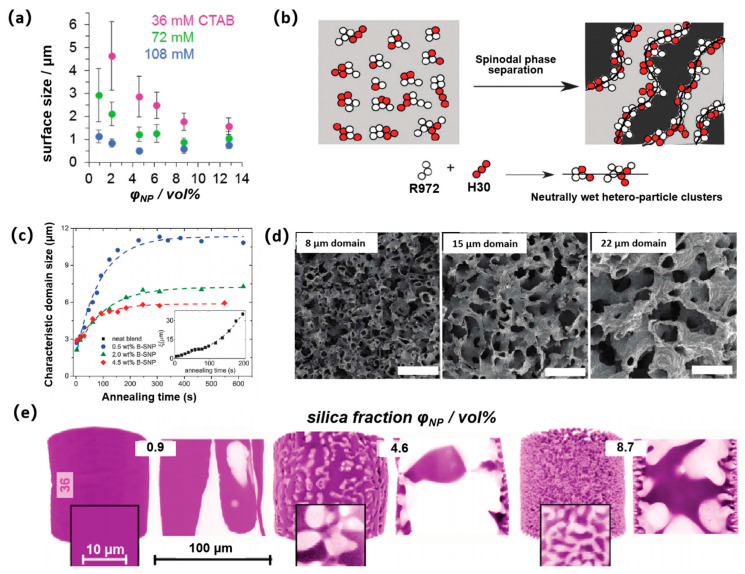
Tunability of Bijel structures. (**a**) Surface pore size of the fibers as a function of the volume fraction of particles (*φ*_NP_) and CTAB concentration (*c*_CTAB_). Reprinted with permission from Ref. [32]. Copyright 2015, Wiley-VCH. (**b**) Illustrating a possible mechanism for interfacial jamming by a mixture of R972 (31% silanol groups) and H30 (50% silanol groups) fumed silica particles. Reprinted with permission from Ref. [66]. Copyright 2015, Royal Society of Chemistry. (**c**) Characteristic domain size (x) of PS/PB/B-SNP Bijels with different weight fraction of particles. Reprinted with permission from Ref. [26]. Copyright 2015, Royal Society of Chemistry. (**d**) The composite electrodes with 8 µm, 15 µm, and 22 µm domains. Reprinted with permission from Ref. [40]. Copyright 2016, Royal Society of Chemistry. (**e**) False colored 3D-reconstructions of fiber segments from confocal z-stacks and corresponding equatorial slices. Reprinted with permission from Ref. [32]. Copyright 2015, Wiley-VCH.

**Figure 3 nanomaterials-14-00574-f003:**
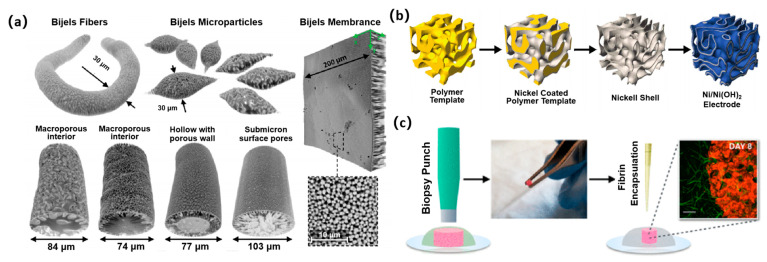
Applications of Bijel-derived porous materials. (**a**) Confocal microscopy images of Bijel fibers, microparticles, and membranes with tunable surface and internal pore structure and size. Reprinted with permission from Ref. [32]. Copyright 2015, Wiley-VCH. (**b**) Fabrication of a Ni/Ni(OH)_2_ composite electrode from a Bijel template. Reprinted with permission from Ref. [40]. Copyright 2016, Royal Society of Chemistry. (**c**) Cell delivery to acellular fibrin schematic. Reprinted with permission from Ref. [74]. Copyright 2018, American Chemical Society.

**Table 1 nanomaterials-14-00574-t001:** A brief summary of key parameter of porous materials prepared by Bijel templating.

Application	Material Morphology	Preparation Methods	Nanoparticle Type	Particle Surface Treatment Method	Particle Size	System	Domain Size	Reference
Bicontinuous porous material	Bicontinuous porous	TIPS	Silica	Drying and Silanization	≈750 nm	2,6-lutidine-water	20–100 µm	[37]
	Microparticles, fibers, and membranes	STRIPS	Silica	CTAB modification	22 nm	diethylphthalate-ethanol-water	500 nm–10 µm	[32]
	Porous scaffold	TIPS	Silica	Drying	410 nm	2,6-lutidine-water	/	[62]
	Bicontinuous macroporous	TIPS	Silica	Silanization	/	1,4-butanediol- poly(ethylene glycol) diacrylate	/	[84]
	Bicontinuous porous	TIPS	Silica	Drying	440 nm	2,6-lutidine-water	6–50 µm	[35]
	Bicontinuous porous	STRIPS	Silica	CTAB modification	22 nm	Hexanediol diacrylate-ethanol-water	<100 µm	[36]
	Porous membranes	STRIPS	Silica	CTAB modification	/	Hexanediol diacrylate-ethanol-water	100–600 nm	[47]
Electrode porous material	Bicontinuous porous	TIPS	Silica	Drying and fluorescently modified	697 nm	2,6-lutidine-water	8–22 µm	[40]
	Bicontinuous porous	TIPS	Silica	Drying and fluorescently modified	/	2,6-lutidine-water	8–15 µm	[77]
	Bicontinuous porous	TIPS	Silica	Drying	/	2,6-lutidine-water	30–85 µm	[85]
Bio-based porous materials	Porous hydrogel	TIPS	Silica	Fluorescently modified	500 nm	2,6-lutidine-water	/	[74]
	Bijels -like hydrogel	Direct mixing	Hydroxyapatite	Arabic gum graft	/	ε-caprolactone-ethanol	/	[39]
	Bicontinuous porous	TIPS	Silica	Fluorescently modified	500 nm	2,6-lutidine-water	/	[86]
	Bijels -like hydrogel	Direct mixing	Gelatin- maltodextrin	Fluorescently modified	/	Gelatin and maltodextrin solutions	/	[87]

## Data Availability

Data sharing is not applicable to this article as no new data were created or analyzed in this study.

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
