# Peer review of "Bicontinuous Interfacially Jammed Emulsion Gels (Bijels): Preparation, Control Strategies, and Derived Porous Materials"

_nanomaterials, 2024, doi:10.3390/nano14070574_

Round 1
Reviewer 1 Report
Comments and Suggestions for Authors
The paper is a review on Bijels. It is well and clearly written and presents most of the aspects, in particular recent knowledge on the subject. This is why it is worth publishing. However, a few minor modifications should be performed:
36 "The idea of stabilizing the bicontinuous liquid-liquid interface with interfacially jammed nanoparticles dates back TO the work of Cates et al. "
136 quenching to (of?) the two-phase region
144 at (on ?) the whole depth of the sample
despite a non volumic heating ?
164 Explain how the co-solvent is removed.
228 Bijels and not Bejels
The font size of figure labels should be the same on all figures. Make it the same on all figures.
302 3.2 size of domain => define what you mean by domain
316-318 "As the particle concentration increased, the morphology of Bijel fibers changed from initial range from separated-voids, to interconnected-void's with finger-like shapes, to a continuous channel extending throughout the fiber" => rephrase, not clear enough
320 replace "Meanwhile" by "Furthermore,"
321-323 "Too low particle concentration is not conducive to the interfacial elasticity for the system to prevent coalescence, resulting in system instability. " => rephrase, not clear enough
334-335 "It was found that Bijels prepared by the TIPS WERE more robust when nanoparticles were used instead of microparticles"
383-384 "the pore size COULD BE large enough to accommodate"
435-436 " facilitated the control OF membrane flux and selectivity"
447-448 "can greatly IMPROVE the efficiency of electrochemical applications" => remove the repeated word
478 "Bijels OFFER a robust, self-assembly based platform for synthesizing" => remove the s at the end of offer
Comments on the Quality of English Language
The English of the paper is of good quality.
Reviewer 2 Report
Comments and Suggestions for Authors
The manuscript of Shen and Cao discusses with the bicontinuous interfacially jammed emulsion gels (BIJELs). Bijels preparation methods (i.e. thermally induced phase separation, TIPS, solvent transfer-induced phase separation, STRIPS, and direct mixing), control strategies (i.e. surface wettability of particles and particle concentration, charge and size) and the derived porous materials (including bicontinuous, electrode and bio-based porous materials) are described in this review. The manuscript is well written and the authors cite appropriate and adequate references. However, some minor comments and suggestions are required to improve this manuscript:
1) In the introduction, provide an example of the interfacial energy calculated for a theoretical particle in order to justify that the particles are stably adsorbed in the interfacial area.
2) In the manuscript, the authors mentioned the vaporization-induced phase separation, the centrifugal and the phase inversion methods have also effective to form Bijels structures. However, the authors developed only thermally induced phase separation, TIPS, solvent transfer-induced phase separation, STRIPS, and direct mixing. I think detailed information about these methods is needed in a review.
3) The figures taken from various publications are difficult to read, for instance, the text in the illustrations are pixelated and difficult to read (see Figures 1, 2 and 3).
4) In the case of degrees of temperature, the International Bureau of Weights and Measures and the U.S. Government Printing Office prescribe printing temperatures with a space between the number and the degree symbol, as in 20 °C. However, it is not respected in this manuscript (see, for instance, page 4, line 143).
Despite these few issues, the manuscript can be published in Nanomaterials after appropriate revisions.
Comments on the Quality of English LanguageThe article is well written and the quality of the English language is decent. Only minor issue: see the 4th point of comments and suggestions for authors.
